# Advancement of Fluorescent and Structural Properties of Bovine Serum Albumin-Gold Bioconjugates in Normal and Heavy Water with pH Conditioning and Ageing

**DOI:** 10.3390/nano12030390

**Published:** 2022-01-25

**Authors:** Bence Fehér, Judith Mihály, Attila Demeter, László Almásy, András Wacha, Zoltán Varga, Imre Varga, Jan Skov Pedersen, Attila Bóta

**Affiliations:** 1Research Centre for Natural Sciences, Institute of Materials and Environmental Chemistry, Magyar Tudósok Körútja 2, 1117 Budapest, Hungary; feher.bence@ek-cer.hu (B.F.); mihaly.judith@ttk.hu (J.M.); wacha.andras@ttk.hu (A.W.); varga.zoltan@ttk.hu (Z.V.); 2Neutron Spectroscopy Department, Centre for Energy Research, Konkoly-Thege M. út 29-33, 1121 Budapest, Hungary; almasy.laszlo@ek-cer.hu; 3Institute of Chemistry, Eötvös Loránd University (ELTE), Pázmány Péter sétány 1/A, 1117 Budapest, Hungary; varga.imre@ttk.elte.hu; 4Department of Chemistry and Interdisciplinary Nanoscience Center (iNANO), Aarhus University, Gustav Wieds Vej 14, 8000 Aarhus C, Denmark; jsp@chem.au.dk

**Keywords:** red-fluorescence, protein–gold conjugates, change in protein conformation, fluorescence, small angle X-ray scattering, small angle neutron scattering

## Abstract

The red-emitting fluorescent properties of bovine serum albumin (BSA)–gold conjugates are commonly attributed to gold nanoclusters formed by metallic and ionized gold atoms, stabilized by the protein. Others argue that red fluorescence originates from gold cation–protein complexes instead, not gold nanoclusters. Our fluorescence and infrared spectroscopy, neutron, and X-ray small-angle scattering measurements show that the fluorescence and structural behavior of BSA–Au conjugates are different in normal and heavy water, strengthening the argument for the existence of loose ionic gold–protein complexes. The quantum yield for red-emitting luminescence is higher in heavy water (3.5%) than normal water (2.4%), emphasizing the impact of hydration effects. Changes in red luminescence are associated with the perturbations of BSA conformations and alterations to interatomic gold–sulfur and gold–oxygen interactions. The relative alignment of domains I and II, II and III, III and IV of BSA, determined from small-angle scattering measurements, indicate a loose (“expanded-like”) structure at pH 12 (pD ~12); by contrast, at pH 7 (pD ~7), a more regular formation appears with an increased distance between the I and II domains, suggesting the localization of gold atoms in these regions.

## 1. Introduction 

Biocompatible red fluorescent BSA–gold conjugates offer favorable biological applications because their red fluorescence significantly differs from tissue materials’ blue or green auto-fluorescence [1]. An elegant and simple one-pot aqueous synthesis of fluorescent bovine serum albumin–gold (BSA–Au) complexes was first described by Xie et al., a pioneering work in the field [2]. In the presence of hydrogen tetrachloroaurate (HAuCl_4_), the interaction between the BSA and gold salt is driven in the alkaline medium (pH ~12), resulting in a significant red luminescence at around 670 nm, which slightly increases when the system neutralizes. Despite intensive studies, there is no consensus about the origin of red emittance. The question is still open—what are the red-emitting parts of the BSA–Au conjugates, and what do they look like? After the inventors, it has been widely accepted that the fluorescent properties result from gold nanoclusters composed of 25 atoms and stabilized by BSA, which were already known to show red fluorescence [2]. It was reported that nanoclusters with 25 atoms have a core-shell structure consisting of an icosahedral core of 12–13 metallic gold atoms surrounded by six Au_2_(SR)_3_ staples covalently bonded to a BSA molecule via the sulfuric groups [2,3,4,5,6,7]. This structural explanation was deduced from matrix-assisted laser desorption-ionization time-of-flight (MALDI-TOF) and X-ray photoelectron spectroscopy (XPS) methods where, unfortunately, water, as a characteristic structural element, is expelled from the system. Other methods, such as Nuclear Magnetic Resonance (NMR), infrared spectroscopy (IR), and small-angle X-ray scattering (SAXS), show that BSA is a dynamic, “living” system assuming different conformations reversibly in an extensive pH range. The aqueous solution of BSA has five different conformations as the function of pH: expanded, fast, normal, basic, and aged forms [8]. In the presence of HAuCl_4_, pH plays an important role. At pH = 12, the BSA-molecules become negatively charged and undergo conformational changes while attracting Au(III) ions, predominantly in regions rich in reducing tyrosine and histidine residues. The key to understanding the red emission’s origin is clarifying the interactions between the neighboring Au atoms or ions [9]. Dixon and Egussa observed that these gold cation–protein complexes caused their red emissions after a further reduction process using sodium borohydride, whereby metal nanoparticles were formed. Therefore, they supposed that the BSA–gold compounds, described by Xie et al. as Au_25_ nanoclusters, were BSA–cationic gold complexes [10]. Moreover, they observed that conformational changes of BSA influence the fluorescence behavior of complexes. They also suggested that the origin of the red fluorescence involves an energy transfer among chromophores formed by the gold and protein residues. The same authors concluded that cysteine, 34 of which form disulfide bonds in BSA, is the binding site of Au(III) but not the location of the red-emitting fluorophore. Recently, the same research group identified the Au(III) binding domain of BSA and localized the origin of red fluorescence within the N-terminal domain using limited proteolysis and molecular cloning techniques based on luminescence measurements [11]. However, the changes and evolution in fluorescent properties are primarily connected to global changes in BSA–gold complexes dependent on the pH value and accompanied by significant structural changes on the atomic level as observed by small-angle X-ray scattering and infrared spectroscopy [12]. The conformation of BSA is not reversible after a neutral–alkali–neutral cycle, and its consequences in fluorescence can be observed. The irreversibility is more pronounced in the presence of Au(III) ions, indicating the importance of hydration effects. Besides the advantages of heavy water use in the neutron scattering techniques, it provides a solvent milieu different from normal water, enabling the observation of hydration-induced structural and conformational changes. It is well-known that the deuterium–hydrogen isotope effect causes significant changes in the folding–unfolding processes of proteins [13,14]. In this work, we show that heavy water, compared to normal water, induces more decided effects in both global and fine structures and that these changes bring a significantly increased red fluorescence than that observed in normal water.

## 2. Materials and Methods

### 2.1. Materials

Bovine serum albumin (BSA, >99%) and HAuCl_4_·3H_2_O (99.99%) were purchased from Sigma-Aldrich (Sigma-Aldrich, St. Louis, MO, USA) and used as received. The pH of the solutions was adjusted with HCl (Sigma-Aldrich, St. Louis, MO, USA) and NaOH (Sigma-Aldrich, St. Louis, MO, USA). All solutions were prepared in ultra-pure Milli-Q water (total organic content ≤ 4 ppb; resistivity ≥ 18 MΩcm) and heavy water (deuterium oxid for NMR, 99.8% D) purchased from Acros Organics (Morris Plains, NJ, USA).

### 2.2. Sample Preparation

A total of 1.67 w% BSA–Au(III) stock solutions in normal and heavy water were prepared by mixing 2.5 w% BSA solutions with 10 mM HAuCl_4_ solutions in 2:1 volume ratio (corresponding with 13:1 ion–protein molar ratio) at moderate stirring rate (600 rpm). After mixing, the pH of the stock solutions was set to 12 (pD to ≈11.6) by dilute NaOH solutions (1 M, both in normal and heavy water) under stirring. After storage at room temperature for two days, the BSA−Au(III) systems were heat-treated at 37 °C for 2 h, then neutralized and stored at room temperature. The beginning and transitional states at pH = 12 (pD ≈ 11.6) and neutral pH (pD) were measured with fluorescence, infrared spectroscopy, and DLS methods. The same preparation protocol was used for the X-ray and neutron scattering methods.

### 2.3. Fluorescence Spectroscopy 

The luminescence of the BSA–Au conjugates was measured using a Jasco FP8500 spectrofluorometer (Jasco International Co., Ltd., Tokyo, Japan) at 25 °C in MQ water and 360 ± 5 nm excitation in the 380–750 nm range. The spectral correction function was assessed by the Maroncelly dye setup [15]. The luminescence quantum yields were determined relative to the 0.546 value of quinine sulfate in 1N sulfuric acid [16]. It must be noted that considering the low sensitivity of the fluorometer in the red region, the red band’s maxima, and the corresponding fluorescence yields, may be slightly underestimated. The air-saturated samples were measured in a 3 mm × 3 mm × 40 mm quartz cuvette with an optical density at the excitation wavelength around 1.4.

### 2.4. Infrared Spectroscopy 

Attenuated total reflection-Fourier transform infrared (ATR-FTIR) spectroscopy measurements were conducted using a Varian 2000 FTIR Scimitar spectrometer (Varian Inc., Palo Alto, CA, USA) fitted with a liquid nitrogen-cooled mercury cadmium telluride (MCT) detector and a ‘Golden Gate’ single reflection diamond ATR accessory (Specac Ltd., Orpington, UK). A sample amount of 5 µL was dropped onto the diamond ATR surface, and the dry film spectrum was collected (at 2 cm^−1^ resolution and 64 scans) after the slow evaporation of the solvent under ambient conditions. Each data acquisition was followed by ATR correction. Spectral deconvolution was performed using the GRAMS/AI (7.02) spectroscopy software (Thermo Galactic, Walthman, MA, USA). Band positions for curve fitting were established using the second derivative and were fixed during the fitting procedure. Band shapes were approximated by Lorentzian functions. The intensities and the bandwidth of each component were allowed to vary until the minimal *χ*^2^ parameter was reached. After the fitting procedure, the relative contribution of a particular band component was calculated from the integrated areas of the individual components [17].

### 2.5. Dynamic Light Scattering

Dynamic light scattering (DLS) of the samples was measured at 20 °C using a W130i dynamic light scattering device (Avid Nano Ltd., High Wycombe, UK) with a diode laser (660 nm) and a photodiode detector. Eppendorf disposable cuvettes (50–2000 µL, UVette routine pack, Vienna, Austria) with a 1 cm path-length were used [18]. Samples containing approx. 10 µM BSA were measured in a final volume of 80 µL in MQ water. We measured the time-dependent autocorrelation function for 10 s, repeated it ten times, and reported the average distributions. A data analysis yielding the mean hydrodynamic diameter (D_h_) and polydispersity (%) was performed with iSize 3.0 software supplied with the device. 

### 2.6. Small-Angle X-ray Scattering

Small-angle X-ray scattering measurements were performed using CREDO, an in-house transmission geometry setup [19,20]. Thin-walled quartz capillaries with a 1.5 mm average outer diameter were filled with samples. After proper sealing, these were placed into a temperature-controlled aluminum block inserted into the vacuum space of the sample chamber. Measurements were recorded using monochromatized and collimated Cu Kα radiation (1.542 Å wavelength); the scattering pattern was recorded in the range of 0.02–0.5 Å^−1^ in terms of the scattering variable/momentum transfer (*q* = (4π/*λ*) sin *θ*, where *2θ* is the scattering angle, and *λ* is the X-ray wavelength). The total measurement time was 7.5 h for each sample. In order to assess sample and instrument stability during the experiment, the exposures were recorded in 5-min units, with frequent sample change and reference measurements. These individual exposures were corrected for beam flux, geometric effects, sample self-absorption, and instrumental background, as well as calibrated into physical units of momentum transfer and volume-normalized differential scattering cross-sections (absolute intensity, cm^−1^ × sr^−1^). The corrected and calibrated 5-min scattering patterns were azimuthally averaged to yield a single one-dimensional scattering curve for each sample.

### 2.7. Small-Angle Neutron Scattering

Small-angle neutron scattering measurements were recorded with the “*Yellow Submarine*” diffractometer operating at the Budapest Neutron Centre [21,22]. Two sample-detector distances of 1.2 and 5.3 m and a quasi-monochromatic neutron wavelength of 0.42 nm allowed us to cover a *q*-range of 0.01–0.5 Å^−1^. Liquid samples were filled in quartz cells with a 2 mm path length, and the measurements were recorded at 25 °C. The raw data were corrected for sample transmission, cell and room background scattering, and the absolute intensity scale was calibrated by the level of incoherent scattering from an H_2_O sample. 

## 3. Results and Discussion

### 3.1. Red Emission Characterized with Fluorescence Spectroscopy

The luminescence spectra of BSA–Au conjugates were studied in normal and heavy water solutions. Two hours after mixing the two basic (BSA and HAuCl_4_) solutions (at pH = 12 and pD ≈ 11.6), luminescence appears in both normal and heavy water systems. The emission, however, is very low in terms of quantum yield, and the moderate deviation of the emission spectra indicates different characteristics for the two systems, as shown in Figure 1A. Indeed, a simple visual observation of the systems, with a blue laser pointer, already indicates the rapid evolution of red emission in H_2_O, whereas the amount of time required for D_2_O is more prolonged, at least one day. Blue luminescence (centered at 440 nm) appears in both systems but is relatively larger in normal water. Two days after setting the pH (pD) to neutral (performed two days after the preparation at alkaline conditions), the systems show drastic changes in their luminescence spectra (Figure 1B). The red emission becomes intensive and turns into the prevalent range. The previously observed blue fluorescence is also present but is not the characteristic feature anymore. Interestingly, the red emittance of the D_2_O system is significantly higher than H_2_O. These differences reflect in the fluorescence quantum yield values, which are 0.035 in the presence of D_2_O and 0.024 in the H_2_O system.

All spectra in Figure 1 can be described by three distinct bands, with the maxima slightly blue-shifted when the deuterated solvent was used. One may conclude that at pH (pD) ≈ 7, after longer conditioning the samples at pH (pD) ≈ 12 (shown in Figure 1B), the fluorescence yields increase by a factor of ten, while the red emission bands become much more intensive. The maximum of the lowest energy band is shifted to red with conditioning, especially for D_2_O (requiring longer relaxation time). In the case of the conditioned samples, the deuteration of the solvent results in a 50% increase in fluorescence yield, partly resulting from the broadening of the red band. The concomitant blue and red luminescence intensities were followed through several days. The ageing time-course of the red luminescence showed drastically different characters in the two aqueous systems. The red luminescence showed high intensity in normal water after 2 d; then, the values reduced significantly. However, in D_2_O, the increase in red luminescence took several days, indicating a longer formation time for the more effective red emission with the concomitant configuration of BSA–Au conjugates. 

### 3.2. Fine Structural Perturbations Observed by Infrared Spectroscopy

Possible conformational changes of BSA, following preparation steps and luminescence development, were inspected by ATR-FTIR spectroscopy. We focused on the amide I and amide II band regions (from 1750 to 1350 cm^−1^), belonging to the C=O, and N–H and N–C vibrations of peptide bonds from protein backbones, respectively, as shown in Figure 2A.

Amide I, composed mainly from the C=O vibration of the peptide bonds, is sensitive to the H-bonding network of the protein backbone. Consequently, by band deconvolution [23,24], the secondary structure of proteins can be deduced (Table 1). At pH = 12, the BSA structure in the BSA–Au bioconjugate is dominated by random coils and β-sheets, resulting in a broad amide I band with a peak at 1647 cm^−1^. These results align with our previous observation at pH = 12, when the BSA backbone became extended with loose unordered parts [12]. The multiple-band structure of amide II also confirms the open, elongated protein geometry with exposed COO^−^ groups of amino acid side chains (affirmed by the shoulders at 1571, 1502 cm^−1^, and the band around 1404 cm^−1^). Compared to the spectrum of the native BSA [12], however, the relative increase in intensity suggests that Au ions may interact with the exposed, negatively charged carboxy groups of the elongated protein.

After 1 d, a slight “reorganization” of protein structure can be observed, reflected by the amide I peak shift toward a higher wavenumber (from 1647 to 1650 cm^−1^). However, more significant spectral changes are witnessed upon adjusting the pH to neutral (pH = 7). The amide I band peak is shifted toward 1654 cm^−1^, presuming a dominantly helical structure. The shoulder bands in the amide II region, assigned to COO^−^ groups of the amino acid side-chains also decrease. We assumed that protein refolding in BSA–Au conjugates, forced by H-bonds formation upon pH adjustment, might lead to the development of Au–Au interactions. After 2 d, a “relaxation” occurs in the amide I peak, but no changes occur in the amide II and side-chain band features. A detailed analysis after band deconvolution also revealed the presence of intermolecular β-sheets suggesting that Au association occurs in the final state at pH = 7 (Table 1).

Using D_2_O as a solvent, due to the hydrogen/deuterium exchange, the IR spectra of BSA–Au systems are different (Figure 2B). The amide I band of unordered protein conformations appears at 1642 cm^−1^. The amide II band (N–H deformation vibration of peptide bonds) at 1538 cm^−1^ is suppressed, and a new amide II’ band (N–D deformation vibration of peptide bonds) at 1431 cm^−1^ is raised. It is worth noting that the bands of exposed charged carboxylate groups (at 1576 and 1499 cm^−1^) are also observable. After 1 day, another slight ‘relaxation’ can be noticed; however, now the amide I’ peak is shifted in the opposite direction, toward a lower wavenumber. It appears that in D_2_O, disordered or sheet-like protein structures are formed in the BSA–Au conjugate. By adjusting the pH (pD) to 7, the alteration in spectral features resembles the BSA–Au/H_2_O system with a stronger contribution of helical conformations. After 2 d, however, the sheet-like or disordered structure is favored (Table 2).

In conclusion, it seems plausible that in alkaline environments, the initial BSA–Au interaction is also affected by the choice of solvent and influenced by the BSA’s geometry. In a neutral state (pH = 7 and at pD ~7), a helical structure tends to form to different extents, resulting in slightly different protein secondary structures. This finding aligns with the significant fluorescence changes of BSA–Au conjugates. 

### 3.3. Conformational Changes of BSA–Au Conjugates Observed by Small-Angle X-ray and Neutron Scattering

To receive insight into the global structure of the conjugate prepared in D_2_O, with special emphasis on the BSA, we performed small-angle X-ray and neutron scattering experiments. The X-ray scattering length density (SLD) is proportional to the atomic number, whereas the neutron SLD depends on the neutron scattering cross-section of nuclei. Thus, SANS and SAXS provide slightly different information on the two systems; however, the scattering is likely dominated by protein scattering in both cases. Due to the lesser amount of gold ions compared with proteins (13:1 Au ion to BSA molar ratio), the scattering contribution of Au was expected to be negligible, even if concentrated in small nanoclusters.

In Figure 3, the one-dimensional SAXS and SANS curves are presented. The radii of gyration were determined by both the Guinier approximation and indirect Fourier transformation (IFT) and are presented in Table 3 [25,26]. The forward scattering was also determined by IFT. The radius of gyration for BSA in D_2_O is in good agreement with the value of BSA in normal water [27]. Adding gold salt did not change this result significantly (pD ~12, SANS result). However, adjusting the pD to 7 results in an increased radius of gyration around 39 Å, which coincides with the literature value for the BSA dimer [27]. SAXS yields slightly larger values than SANS for the same pD states. The reason for this lies in differences between the X-ray and neutron techniques and the uncertainty of R_G_ determination. The forward scattering of samples at pD ~7 is approximately double that of pD ~12 for both SANS and SAXS, as shown in Figure 3 and Table 3. Since forward scattering is proportional to the molar mass of the scattering objects, we concluded that dimerization occurs during the neutralization because the concentration of the samples is the same. The higher SAXS intensity (compared to SANS) results from the different scattering mechanisms. We noted that dimerization did not occur in normal water, which is a significant difference in the formation of BSA–Au conjugates in the two water systems [12]. Figure 4 presents the Kratky plots of the SAXS curves (taken on D_2_O systems). This representation emphasizes the high-*q* region of the scattering curves, which is only reliable for SAXS because the incoherent scattering of SANS experiments renders the background subtraction slightly imprecise. The sample curve at pD ~12 increases with high *q*, which is characteristic to (at least partially) unfolded proteins. However, at pD ~7, the Kratky plot exhibits a maximum at low-*q* and plateau at high-*q,* indicating that the protein has folded. 

To reveal protein structural changes in the complex system caused by pH variation, we performed ab initio modeling with DAMMIF and rigid-body refinement with SASREF on curves obtained by SAXS and SANS [28,29]. The native BSA structure was grouped into four linked domains (marked by different colors in Figure 5 and Figure 6: domain I: residue 1–147 (blue), domain II: residue 148–300 (magenta), domain III: residue 301–495 (green), and domain IV: residue 496–583 (orange). Ten runs were performed with DAMMIF and SASREF, and the obtained structures were processed by DAMAVER [30]. For samples with pH (or pD) adjusted to 7 and the presumed dimer structures, a P2 symmetry was imposed. Additionally, the dimer structure was also fitted without requiring symmetry, but the fit did not improve significantly. Therefore, we chose to keep the requirement on the P2 symmetry since it decreases the degrees of freedom and yields more robust results.

The scattering curves of SASREF fits are shown along with measured ones in the graphs on the left-hand side of Figure 6, while the corresponding models obtained by both DAMMIF and SASREF on the right-hand side are aligned and overlaid. The resulting structures from the two different simulation methods are in good agreement. As a reference, the BSA in D_2_O without HAuCl_4_ was also fitted. The structure of BSA in D_2_O resembles normal water, as observed in our previous work [12]. From the SASREF fits of SANS curves, we concluded that the four-domain structure satisfactorily describes the scattering data (χ^2^ values are presented in Table 3), suggesting that the conformation of the protein is not seriously affected by the gold salt. However, a relative change in the domains’ arrangement can be observed by adding HAuCl_4_ and increasing the pD to ~12. Adjusting the pD to ~7 allows for further change in the domains’ arrangement besides dimerization. To receive insight into the motion of domains, we defined the ‘main axis’ of each domain by the first and last amino acid of the domain. Then, the angles of two adjacent domains were calculated according to their main axis for each sample in all 10 SASREF simulation runs, averaged, and normalized to the appropriate angles of BSA in D_2_O without HauCl_4_. The changes in the domains’ arrangement were similar in normal and heavy water. The results presented in Figure 7 show that the relative position of domain I and II at pD 12 changes significantly compared with pure BSA at pD 7. On the other hand, adjusting the BSA–Au system’s pD value to ~7, and increasing the angles of two adjacent domains, similar to an opening, is further enhanced. No clear trend was observed in the relative positions of domains II, III, and IV. These results suggest that the domain rich in Au(III) (or Au) is localized between domains I and II. This finding is in good agreement with the recently published work of Dixon et al., who claimed that the fluorophore is located on the fragment of protein from residue 115–312 (almost perfectly covered by domain II in our case) based on proteolysis of the BSA–gold conjugate.

### 3.4. Dynamic Light Scattering

Apart from some methodological uncertainties and unknown diluting effects inevitable by measuring, dynamic light scattering also shows differences in the solvation effect between normal and heavy water. Although the BSA parent system is not the main subject of our present work, it is reasonable to recall its apparent size difference in the two solvents. In normal water, native BSA shows a small mean hydrodynamic diameter (approx. 37 Å) at pH = 7, extending to a significantly larger value (approx. 99 Å) at pH = 12 due to unfolding. These diameters show similar values in heavy water (38 and 81 Å, respectively). In the presence of Au(III) ions in water, the characteristic value decreases from 131 Å to 115 Å if the pH value changes from 12 to 7. Hydrodynamic diameters show drastically different tendencies in heavy water as the value increases from 126 Å to 233 Å upon neutralization (Table 4). We observed that the latter value was practically halved (119 Å with 70.7% polydispersity) after a mild ultrasonic treatment, indicating the existence of BSA–Au conjugate dimers which can fall apart in the D_2_O matrix.

Besides the good agreement with model calculations (presented in Figure 5 and Figure 6), comparing radii of gyration to the mean hydrodynamic diameters also provides the same conclusion. One must consider that SAXS and SANS measure the “compact cores” of conjugates uninfluenced by the hydration layer. Radii of gyration increased by about 40% after neutralization in D_2_O (Table 4). A decrease was observed in a previous study from 44.7 Å to 37.5 Å in H_2_O, which aligns with our present DLS data and strengthens the fact that dimerization does not occur in the normal water system. The fluorescence measurements met this finding since the red emission spectrum in the D_2_O system is wider than that in H_2_O, emphasizing structural differences in the two systems. Supposedly, the extended dimer conjugates contribute to energy transfer mechanisms and, finally, increase the quantum yield in heavy water.

## 4. Conclusions

Our experimental studies revealed that red fluorescence’s evolution is not exclusively related to protein–gold interactions and is strongly influenced by hydration effects. The perturbation differences in hydration processes induced by the hydrogen–deuterium isotope effect were revealed by fluorescence and infrared spectroscopies. The fine structural changes observed were manifested in the luminescence behavior. Since the conformation of the protein is highly affected by the (actual) H(D)-bond network, the medium appears to play an important role in the conformation-related optical properties. Indeed, the fluorescence quantum yield of the BSA–Au conjugates in heavy water increased significantly (50%) compared with normal water-containing matrix. Collaterally, the alterations of fine atomic structures observed with IR spectroscopy are in rapport with the conformational changes appearing at significantly larger dimensions. The use of a single rigid-body model is admittedly a crude one; however, the changes in the scattering curves induced by choice of medium (H_2_O or D_2_O), as well as the pH (or pD), were found to be much larger than the effects of conformational dynamics of the protein, justifying the use of a single, representative conformational model for the dynamic ensemble in all cases [31].

The choice of medium slightly affects the overall protein conformation. The trend of domain motion upon adjusting the pH in alkaline and neutral intervals is the same in both water systems, with only the first two domains affected by the interaction with Au. Dimer formation in BSA–gold conjugates at pH = 7 and even pD ~7 was observed with scattering techniques. The connected fine structural, configurational, and optical features may indicate that the developed BSA–gold ion bioconjugates (sensitive to several kinds of external perturbations) are similar to associates of polyelectrolyte and gold ions, rather than “robust” compact nanoclusters attached to the BSA protein. The versatility of the occurring energy transfer in these associates may explain the observed complex luminescence features.

## Figures and Tables

**Figure 1 nanomaterials-12-00390-f001:**
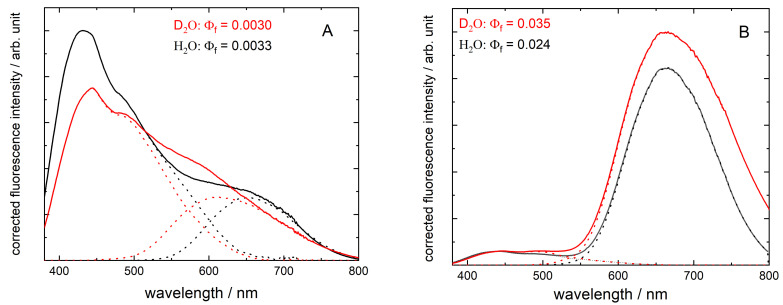
Corrected fluorescence spectra of BSA–Au conjugates in normal (black lines) and heavy water (red lines). In (**A**), the arbitrary intensities are multiplied by a factor of ten, compared with (**B**), the luminescence was detected just 2 h after reaching pH (pD) = 12, whereas in (**B**), after conditioning the samples at pH (pD) = 12 for 2 d, and subsequently neutralizing to pH = 7 (pD ≈ 7) (2 d). (The dotted lines indicate an approximate resolution of the reddest band and the other two bands together.).

**Figure 2 nanomaterials-12-00390-f002:**
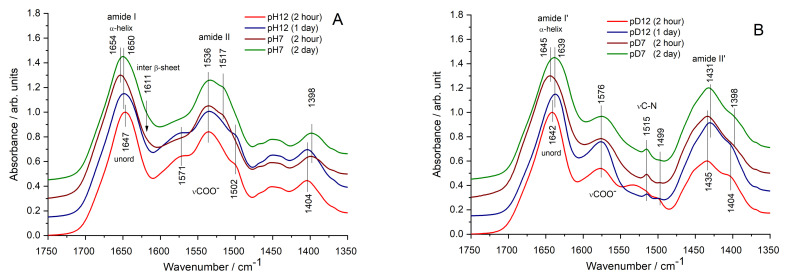
Amide I and amide II regions of IR spectra obtained from BSA–AU bioconjugate samples in H_2_O (**A**) and in D_2_O (**B**). Spectra are normalized to the highest amide I peak and shifted vertically for improved visualization.

**Figure 3 nanomaterials-12-00390-f003:**
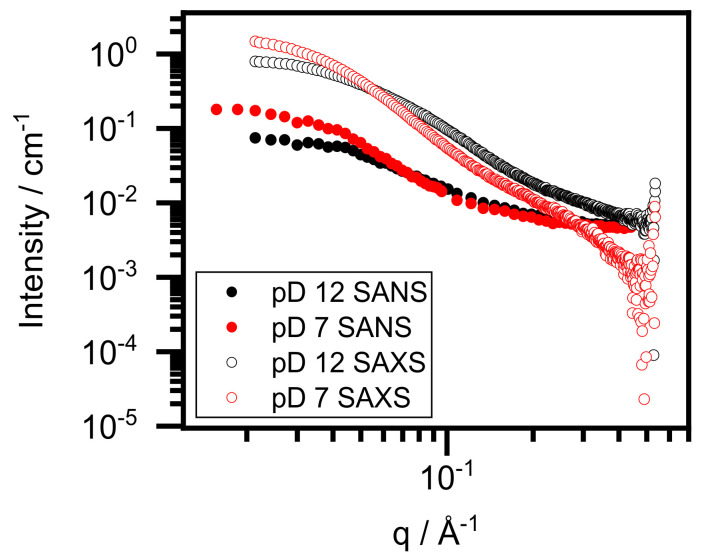
Small-angle X-ray and small-angle neutron scattering curves of BSA–Au conjugates at pD ~12 and pD ~7.

**Figure 4 nanomaterials-12-00390-f004:**
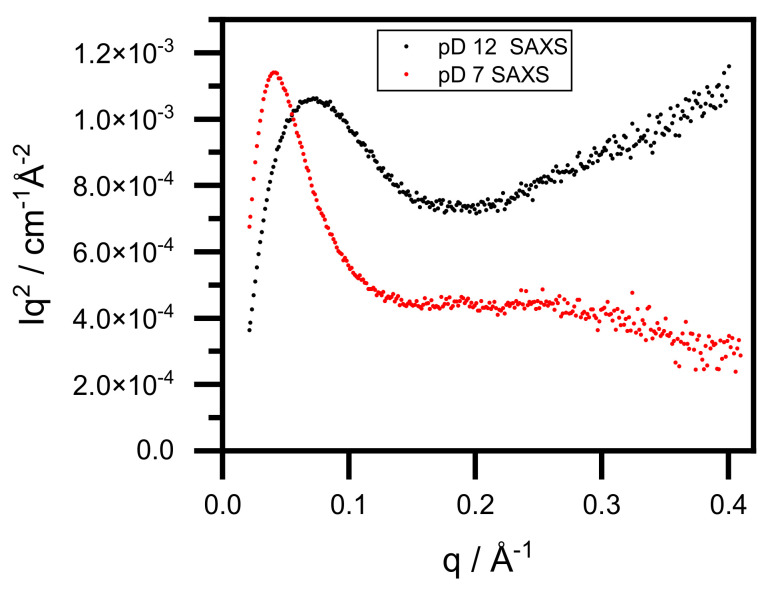
Kratky representation of small-angle X-ray scattering curves of BSA–Au conjugates at pD ~12 and pD ~7.

**Figure 5 nanomaterials-12-00390-f005:**
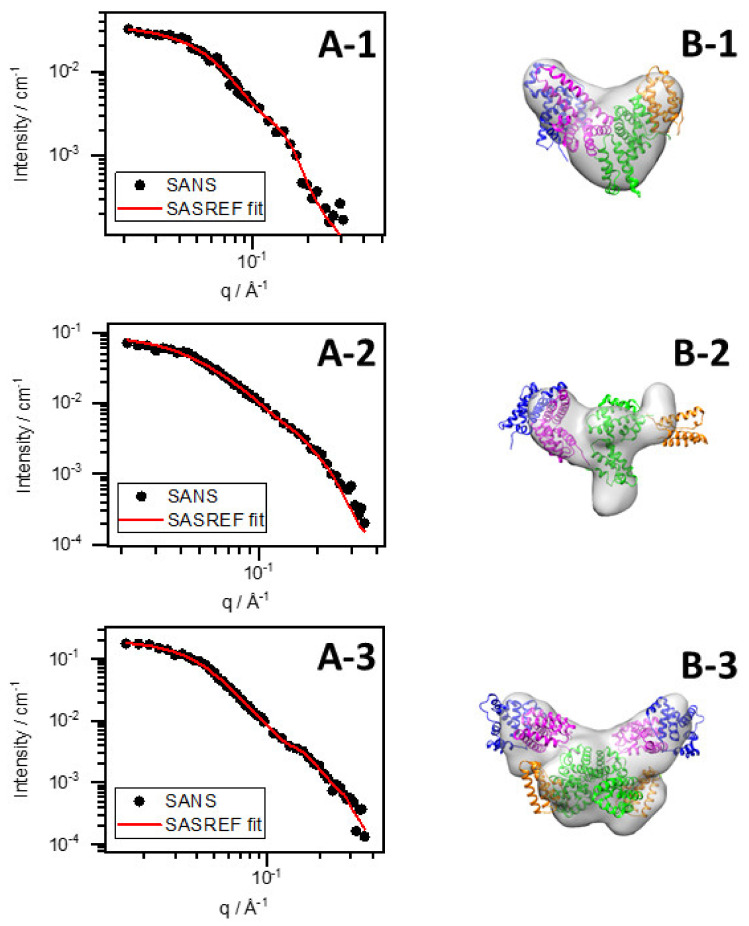
SANS curve of native BSA at pD ~7 and SASREF fit (**A-1**), SANS curve of BSA–Au conjugates at pD ~12 and SASREF fit (**A-2**), SANS curve of BSA–Au at pD ~7 and SASREF fit (**A-3**), obtained SASREF (cartoon) and DAMMIF (surface) structures (**B-1**–**B-3**).

**Figure 6 nanomaterials-12-00390-f006:**
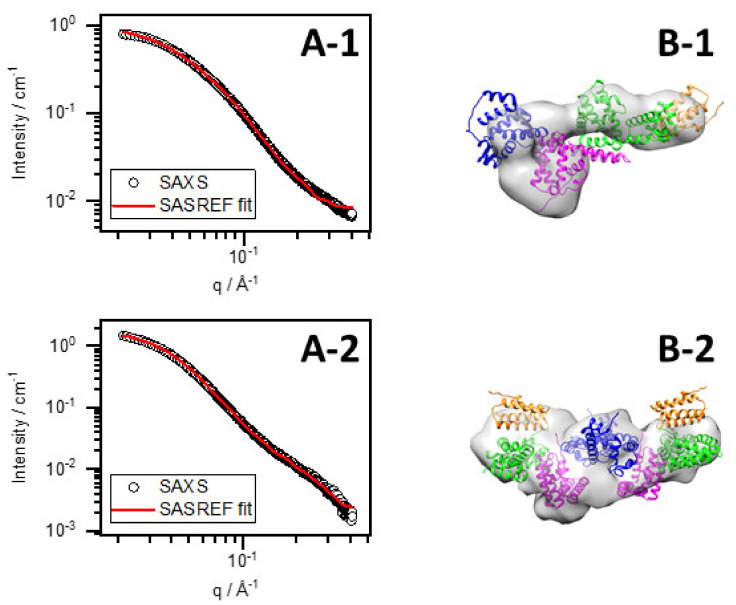
SAXS curve of BSA–Au conjugates at pD ~12 and SASREF fit (**A-1**), SAXS curve of BSA–Au conjugates at pD ~7 and SASREF fit (**A-2**), obtained SASREF (cartoon) and DAMMIF (surface) structures (**B-1**,**B-2**).

**Figure 7 nanomaterials-12-00390-f007:**
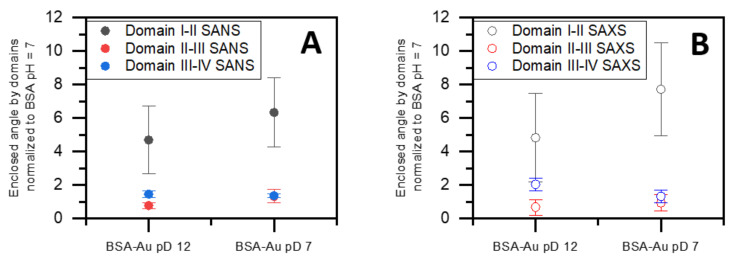
Enclosed angle by domain I–II, II–III, and III–IV obtained by SASREF from SANS (**A**) and SAXS (**B**) curves.

**Table 1 nanomaterials-12-00390-t001:** Secondary structure of BSA–Au conjugate system in H_2_O estimated by the deconvolution of amide I IR band (expressed in %, the peak position of the component bands is provided).

	β-Turns 1687 cm^−1^	α-Helix 1656 m^−1^	Loose α-Helix 1651 cm^−1^	Random Coil 1648 cm^−1^	β-Sheet 1638 cm^−1^	Inter. β-Sheet 1611 cm^−1^
pH12	17	-		74	9	-
pH12 (2 d)	35		23	-	42	-
pH7 (2 h)	32	33		-	33	2
pH7 (2 d)	27	-	41	-	29	3

**Table 2 nanomaterials-12-00390-t002:** Secondary structure of BSA–Au conjugate system in D_2_O estimated by deconvolution of amide I IR band (expressed in %, with the peak position of the component bands).

	β-Turns 1687 cm^−1^	α-Helix 1656 cm^−1^	Loose α-Helix 1651 cm^−1^	Random Coil 1648 cm^−1^	β-Sheet 1638 cm^−1^	Inter. β-Sheet 1611 cm^−1^
pD12	21	-	-	70	-	9
pD12 (2 d)	35	-	-	-	62	3
pD7 (2 h)	29	1		-	70	-
pD7 (2 d)	34		15	-	51	-

**Table 3 nanomaterials-12-00390-t003:** Parameters obtained by the data evaluation of SANS and SAXS curves.

	BSA pD7 SANS		BSA–Au pD12 SANS		BSA–Au pD7 SANS		BSA–Au pD12 SAXS		BSA–Au pD7 SAXS	
I(0) (cm^−1^)	0.037 ± 1 × 10^−3^		0.076 ± 1 × 10^−3^		0.200 ± 4 × 10^−3^		0.948 ± 1 × 10^−3^		1.970 ± 0.026	
RG (Å) Guinier fit	28.4 ± 0.9		27.1 ± 0.1		38.3 ± 0.4		31.60 ± 0.06		43.0 ± 0.390	
RG (Å) IFT	29.75 ± 1.12		28.47 ± 0.40		39.37 ± 0.46		33.71 ± 0.04		44.98 ± 0.30	
Fitting method	DAMMIF	SASREF	DAMMIF	SASREF	DAMMIF	SASREF	DAMMIF	SASREF	DAMMIF	SASREF
χ2	1.11	1.09	1.09	1.95	1.63	1.04	1.95	6.56	0.73	1.46

**Table 4 nanomaterials-12-00390-t004:** Hydrodynamic radius of the BSA–Au and BSA systems in H_2_O and D_2_O at given pH (pD) values. The polydispersity index is also given in %.

	BSA pH7	BSA pH12	BSA–Au pH12	BSA–Au pH7	BSA pD7	BSA pD12	BSA–Au pD12	BSA–Au pD7
Diameter, D_h_ (Å)	37	99	131	115	38	81	126	233/119 *
Polydisp. (%)	57.7	16.7	23.8	27.5	41.0	19.9	21.5	15.9/70.7 *

* after ultrasonic treatment.

## Data Availability

Processed and derived data are available from the first author B.F. on request.

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
