# Peer review of "Advancement of Fluorescent and Structural Properties of Bovine Serum Albumin-Gold Bioconjugates in Normal and Heavy Water with pH Conditioning and Ageing"

_nanomaterials, 2022, doi:10.3390/nano12030390_

Round 1
Reviewer 1 Report
Please find the review report in the attachment.

Author Response
Responses to the Reviewers
Dear Reviewers, 16/01. 2022
We thank your helpful comments and questions and also your positive endorsements of our manuscript. We try to give adequate responses to each of your major comments below (minor requests, such as incorrect dimensions and typos have been fixed).
Yours sincerely: the authors
Specific Answers to Reviewer #1
1/a. “The measurements in H2O (at least SAXS) were not made?”
The effect of pH on the hydrated BSA system (both in the presence and absence of gold ions) was previously investigated in H2O (B. Fehér et al. J. Mol. Liquids, 2020, 309, 113065), cited and referenced as [12] in the manuscript. In this work, we published the SAXS data. Unfortunately, we were not able to model the state at pH=12 within the frame of the applied software (DAMMIF, SASREF). However, the Kratky plot of the SAXS data obtained in H2O system indicated that the system is unfolded at pH=12. Moreover, the formation of dimers was not observed in H2O at pH=12 and even not at pH=7. These facts are mentioned in the corrected manuscript.
1/b. “…This would be useful e.g. for a comparison with DLS measurements.”
We agree with the opinion of the reviewer. For this purpose, we extended the section of “Dynamic light scattering” and the DLS and RG data were compared to each other. We have also used data from our previously published work ([12]) concerning to the SAXS measurement in H2O.
- “The unit of Guinier radius is Angstrom, not reciprocal Angstrom (Table 3).”
Thank you very much for highlighting this serious mistype, we fixed it.
- “Double I(0) value upon neutralization suggesting dimerization, as argued in the text, is visible in Table 3 for SANS but not for SAXS.”
We thank you very much for observation this error. This was also a typo, which occurred at the construction of the Table (You can see the scattering data in Fig. 3, in which the forward scattering of the SAXS curve of BSA-Au pD7 is I(0)=1.97 cm-1. Instead of I(0)=0.197 ± 0.026, the real value is I(0)= 1.97 ± 0.03 cm-1, which is approximately double of the result obtained for the monomer, thus the original argument is valid.
Reviewer 2 Report
Fehér et al. investigated the red-emitting fluorescent properties of BSA-Au conjugates by inspecting the fluorescence and structural behaviors of pure BSA and its Au conjugates in normal and heavy water at different pH/pD using a series of experiments including fluorescence spectroscopy, IR and DLS, as well as SAS (SAXS and SANS). Overall, this manuscript presents a bunch of valuable data and is well organized. I think it deserves publication after addressing my major concerns on the modeling part.
- As pointed out by the authors, previous experimental data have suggested that “BSA is a dynamic, living system, assuming different conformations reversibly in an extremely wide pH range. The aqueous solution of BSA has five different conformations as the function of pH: expanded, fast, normal, basic and aged forms”. So the conformational space of BSA is diverse with multiple states although whose populations might be dependent on the experimental conditions. However, the authors modeled BSA as a single state using rigid-body refinement with SASREF on SAXS and SANS curves. The χ2 values between the model and the SAS curves are close to 1.0, indicating a perfect fitting. However, is it possible the perfect fitting was a result of overfitting? I would suggest they use a multi-state model to fit with the data like in the reference (org/10.1371/journal.pcbi.1007870), if doable.
- The authors suggested that the sheet-like or disordered structure of BSA is favored at pD=12 after 2 days. However, the models were built as rigid-body domains from the native folded structure.
- The Au may contribute substantially to the SAS curves. However, in the BSA-Au conjugate model, only BSA was modeled from the curves.
- The IR spectra provided valuable information on the secondary structures and the SAS data have important information on the overall shape. Both data should be explained in a self-consistent way and it would be better to build the models by integrating both.
Overall, I like the manuscript but I am having doubts about the quality of the models built from SAS data. I would be happy to see the improvement of this part in the revision.
And a few minor points:
- In Table 3, there are two DAMMIF data for the first column “BSA pD7 D2O SANS”. Should the later one be “SASREF”?
- Line 81, “well-know” --> “well-known”.
- Line 291, “the obtained structures was” --> “.. were”.
- Line 302, “two different simulation method are”-->“…methods..”
Author Response
Dear Reviewers, 16/01. 2022
We thank your helpful comments and questions and also your positive endorsements of our manuscript. We try to give adequate responses to each of your major comments below (minor requests, such as incorrect dimensions and typos have been fixed).
Yours sincerely: the authors
Specific answers to Reviewer #2
- “As pointed out by the authors, previous experimental data have suggested that “BSA is a dynamic, living system, assuming different conformations reversibly in an extremely wide pH range. The aqueous solution of BSA has five different conformations as the function of pH: expanded, fast, normal, basic and aged forms”. So the conformational space of BSA is diverse with multiple states although whose populations might be dependent on the experimental conditions. However, the authors modeled BSA as a single state using rigid-body refinement with SASREF on SAXS and SANS curves. The χ2 values between the model and the SAS curves are close to 1.0, indicating a perfect fitting.”
We fully agree with the reviewer. We have studied the experimental conditions (effect of pH, thermal treatment) on BSA-Au conjugates in H2O in a previous work (cited under [12] in the present manuscript). Allowing only a single conformation and applying the rigid-body refinement with SASREF on SAXS is admittedly a crude one, but the configurational/conformational changes we found in the protein structure were significant and in line with the spectroscopic findings. It must also be mentioned that the model description was not complete, as the unfolded states were not describable. Here, we extended the spectroscopic studies (calibrated fluorescence spectroscopy to determine quantum yields) and used heavy water instead of normal water.
- “However, is it possible the perfect fitting was a result of overfitting? I would suggest they use a multi-state model to fit with the data like in the reference org/10.1371/journal.pcbi.1007870), if doable.”
Thank you very much for your valuable suggestion regarding the modeling described in the reference. We tried to reduce the chance of overfitting in our case by performing at least 10 modeling runs and filtering the outliers, thus obtaining generally 8-9 relevant structures, which were physically reasonable, too. The method used in the paper you suggested (A. H. Larsen et al. PLOS Comp. Biol. 04. 27, 2020) is certainly an interesting and novel approach and worth testing on our system. However, several questions must be addressed. First, a similar treatment of the water-protein interaction might be needed, which in our case must be done probably more critically, as we are dealing with an “isotope effect”, which may have a very complex impact on the formation of conformation accompanied by weak interactions. Secondly, the inclusion of gold is also not trivial. While this approach is much more elegant than our present one, and such an examination can be potentially useful in the future, but partly due to the lack of time, partly because our system is more complicated, it is beyond the scope of this paper. A simpler and more rapid way for checking how a large conformational space contributes to the scattering curve is ensemble optimization (Tria, G., Mertens, H. D. T., Kachala, M. & Svergun, D. I. (2015) Advanced Ensemble Modelling of Flexible Macromolecules using X-ray Solution Scattering. IUCrJ 2, 207-217). In this process, a conformational pool of 10000 members is generated based on the amino acid sequence. Subsequently, the initial pool is processed by a genetic algorithm and representative conformations are found. If the results from this process are not adequate (see reference), the protein is to be treated as a rigid molecule. When the algorithm was attempted on our scattering curves, each was classified as corresponding to rigid molecules.
“The authors suggested that the sheet-like or disordered structure of BSA is favored at pD=12 after 2 days. However, the models were built as rigid-body domains from the native folded structure.”
Thank you very much for highlighting the contradiction between our two statements, which should have been better clarified. FTIR reveals the changes on the atomic scale; however, SAS gives low-resolution information about the BSA shape. The two statements correspond to different size ranges. FT-IR spectroscopy measures the ratios of various secondary structure elements (α-helices, β-sheets, etc.) as the relative number of the participating peptide -bonds. On the other hand, the modeling of SAS curves with the native domains is a good assumption to model the data. Of course, we must keep in mind that the result obtained from SAS is only the relative arrangement of the domains, without any information on the fine atomic changes of protein structure.
“The Au may contribute substantially to the SAS curves. However, in the BSA-Au conjugate model, only BSA was modeled from the curves.”
The X-ray scattering intensity of a particle (be it a protein, a gold nanocluster, or just a single Au ion) is proportional to the square of the number of excess electrons above or below that of the displaced solvent. The number of electrons in a gold atom is admittedly large (compared to the atoms composing water or the protein), but in our system, the number of Au atoms is very small (compared to protein atoms), and even if they are not homogeneously dispersed but form nanoclusters (of at most 25 atoms per the literature), these are much smaller than the proteins themselves. Adding the fact that the protein: Au ion molar ratio is 1:13, the SAXS contribution of gold turns out to be negligible (larger Au nanoparticles have not been observed by us). The scattering of gold may only be relevant at very high values of q, where the protein scattering decays with a power-law function. The modeling is mostly based at the low- and intermediate-q ranges, therefore this should not cause any problem in our case.
In SANS, a similar argument applies but with scattering length instead of electrons. Furthermore, even metallic Au is almost ’invisible’ for neutrons in heavy water, since their scattering length densities are very close to each other.
“The IR spectra provided valuable information on the secondary structures and the SAS data have important information on the overall shape. Both data should be explained in a self-consistent way and it would be better to build the models by integrating both.”
This is actually a very important point. Sadly, integrating IR and SAS data is problematic. Namely, IR spectroscopy yields “global” data, i.e. we have no information about the spatial extension of the changes in secondary structures, while SAXS/SANS cannot resolve fine, atomic details. A common ground where the two methods might be combined would be computer modeling like in the paper you suggested above. However, one has to investigate whether the coarse-grained representation of different secondary structures can carry enough structural contrast in SANS/SAXS curves? We hope to be able to address this problem in our subsequent work in this field.
Overall, I like the manuscript but I am having doubts about the quality of the models built from SAS data. I would be happy to see the improvement of this part in the revision.
And a few minor points:
- In Table 3, there are two DAMMIF data for the first column “BSA pD7 D2O SANS”. Should the later one be “SASREF”? -> corrected
- Line 81, “well-know” --> “well-known” -> corrected.
- Line 291, “the obtained structures was” --> “.. were”. -> corrected
- Line 302, “two different simulation method are”-->“…methods..” -> corrected
Round 2
Reviewer 2 Report
Thanks for the responses. I don't think my concerns on the modeling part have been well addressed but I do realize that this is not an easy job to do in a short time. And I understand that the authors want to leave this problem in their subsequent work. I am looking forward to following it.
I have no further comment.